# Development and Application of a Virtual Reality Biphasic Separator as a Learning System for Industrial Process Control

Francisco Flores-Bungacho [1] , Jonathan Guerrero [1] , Jacqueline Llanos [1,*] , Diego Ortiz-Villalba [1] , Alex Navas [2,3] and Paola Velasco [1]

[1] Electrical and Electronic Department, Universidad de las Fuerzas Armadas ESPE, Sangolquí 171103, Ecuador; fiflores1@espe.edu.ec (F.F.-B.); jmguerrero3@espe.edu.ec (J.G.); ddortiz5@espe.edu.ec (D.O.-V.); pamavesa28@gmail.com (P.V.)
[2] Electrical Engineering Department, University of Chile, Santiago 837-0456, Chile; alex.navas@ug.uchile.cl
[3] The Power Electronics, Machines and Control Group (PEMC), University of Nottingham, Nottingham NG7 2RD, UK
[*] Correspondence: jdllanos1@espe.edu.ec; Tel.: +593-3-2810-206

**Abstract:** In this study, we propose a virtual reality biphasic separator methodology in an immersive industrial environment. It allows the training of students or engineers in process and automatic control. On the other hand, the operating performance of a biphasic separator requires advanced automatic control strategies because this industrial process has multivariable and nonlinear characteristics. In this context, the virtual biphasic separator allows the testing of several control techniques. The methodology, involving the immersive virtualization of the biphasic separator, includes three stages. First, a multivariable mathematical model of the industrial process is obtained. The second stage corresponds to virtualization, in which the 3D modelling of the industrial process is undertaken. Then, the process dynamic is captured by the plant model implemented, in the software Unity. In the third stage, the control strategies are designed. The interaction between the virtual biphasic separator and the control system is implemented using shared variables. Three control strategies are implemented and compared to validate the applicability: a classic control algorithm, namely, the proportional integral derivative (PID) control method, as well as two advanced controllers—a numerical controller and model predictive control (MPC). The results demonstrate the virtual separator's usability regarding the operating performance of the virtual biphasic separator, considering the control techniques implemented.

**Keywords:** virtualization; virtual laboratory; biphasic separator; advanced control; prosses control

## 1. Introduction

One industry that drives the economy of many countries globally is the oil industry. Although the industrial processes of this industry are currently controlled, their operation is not necessarily optimal. The oil industry incorporates several processes; among them, one of the most important is the separation process, which can be biphasic, three-phase, and tetra-phase. The separation of two phases occurs before other stages of separation that make up the process [1]. A biphasic separator is a part of the oil collection process, implemented next to the production wells that extract the hydrocarbon by means of electro-submersible pumps. The product of one or more wells reaches the biphasic separator through a set of valves and pipes called the manifold in order to take data into account for production, record trends, and histories [2].

Industrial processes require a dynamic method of analysis to understand their operation and thus propose efficient control strategies. The academic study and analysis of the operation of industrial processes allow the generation of application solutions at an industrial level, which improves the operation of these processes and guarantees the improvement of production processes through their optimal operation. One way to enhance

the operation of an industrial process is through the design of efficient controllers which consider the actual characteristics and dynamics of the process, since an inadequate design will cause problems and reduce the efficiency of the equipment [3].

A biphasic separator involves a process of separation of fluid and gas, in which the force of gravity causes the fluid droplets to fall from the mixture to the bottom of the container, and the gas is directed to the top of the container. The fluid leaves the container through the fluid discharge valve, which a level controller regulates. The level controller is responsible for operating the discharge valve when it detects changes in the height of the fluid. The gas flows horizontally through the gravity sedimentation section above the fluid. As the gas flows through this section, tiny droplets of fluid dragged into the gas are separated by gravity and fall to the gas–fluid interface. The pressure controller is responsible for activating the pressure control valve when changes are detected in the container, controlling the speed at which the gas leaves the vessel to maintain internal pressure [1].

The separation of components or phases (gas and fluid) aims to process these phases into marketable products or process them for proper environmental waste. The gas is isolated from the fluid part to ensure a stable crude in terms of volatility and pressure, which meets marketing criteria. Therefore, separator modeling has become a point of interest for controller design, fault detection, process optimization, and simulation dynamics [4].

Testing different control techniques on a working separator is impossible because it would mean stopping production, representing a high economic impact on the industry. On the other hand, the academy can contribute with experiments on these processes. However, to have a biphasic separator that is identical to the existing ones in the industry, allowing experiments to be conducted with different control strategies to evaluate the optimal performance of processes, is expensive due to the monetary value of industrial components such as transmitters, controllers, and other necessary equipment [5]. With technological advancements, virtual laboratories have become possible. This can be achieved by replicating the dynamic behavior of the variables of interest of an actual process. Furthermore, new technologies make it possible to use several techniques to estimate the dimensions of oil separators [3]. Laboratories have been developed with science-learning approaches such as the virtual laboratory to support chemical reaction engineering, allowing virtual experiments to solve problems taken from real-life engineering [6]. In the same approach, in [7] an augmented reality oyster learning system for a primary school natural science course was proposed. This proposal enhanced learning about marine knowledge, specifically in relation to oysters. In a university environment in [8], a case study of the use of educational games in virtual reality for the teaching process of industrial engineering tools was proposed. In the tele-rehabilitation context in [9], a telerehabilitation virtual system was proposed, reinforcing the implementation of a group of exercises for patients who had experienced injury or pathology.

Laboratories for control applications have been presented, such as a virtual laboratory of a ball and plate system for two experiments (point stabilization, trajectory tracking) [10] and a virtual laboratory for a quadrotor [11]. Although the previously proposed control systems are complex, they are not applications to real-life industrial processes. In this approach, laboratories have been applied to processes such as the virtual laboratory environment for the control design of a multivariable process for a four-coupled-tanks system that could be used in an educational environment [12,13]. In a similar approach, a FESTO virtual workstation was proposed, similar to the one in a real-life laboratory, allowing the level and temperature control of two tanks. That work presented objects in three dimensions and the use of a control algorithm by means of shared memory [14]. Following that study, several applications and studies have been carried out, such as the evaluation of advanced control strategies that have improved the performance of the variables to be controlled [15]. These virtual process labs have undoubtedly contributed to learning in academia; however, these processes must be oriented towards real-life industrial applications.

Developing virtual processes for the modelling of real-life industrial processes is of interest to industry because they can be used to perform controls that optimize the processes and which, after evaluation, can be implemented. In addition, they can be used in the training of new personnel, knowledge updates, and even to perform tests that cannot be implemented in a real life process, etc. In this context, a virtual laboratory for energy generation based on biodiesel production has been presented, which allowed the study of the basic properties of biofuels, simulating reality step-by-step, enabling research in areas such as biofuel characterization [16]. Furthermore, in relation to the induction of new employees, a study involving an industrial pasteurizing plant is shown in [17], in which a virtual laboratory allowed interaction with this industrial environment in order to understand the stages of the pasteurization process within the dairy industry [17].

Having virtual plants of real-life processes, where personnel can evaluate control strategies that allow the efficient operation of a process without requiring the intervention of the real plant, benefits the industry because it enables offline tests, as well as training and inductions. Therefore, this research paper presents a virtualization methodology of a biphasic separator applied to the oil industry that can be replicated to any other industrial process, which evaluates and compares control strategies considering the nonlinear multivariable characteristics of the process.

Biphasic and three-phase separators have nonlinear and multivariate characteristics that need to be considered in the virtualization and modeling of control techniques for their efficient performance. In this research work, biphasic separators are studied because they are the most economical, they are the most commonly used separators in the local oil industry and because indirectly, through the control of the fluid level variable, they enable the level of oil to be controlled. Their dynamics are analyzed and emulated in a virtual plant, to which different control strategies are applied, to guarantee good performance.

A virtual plant with the real dynamics of a biphasic separator in an immersive environment is beneficial for applications related to the design of control strategies that guarantee the efficient performance of the processes. The control of a biphasic separator has many challenges due to its nonlinear and multivariate characteristics. In fact, many investigations have focused on proposing control strategies for their good performance, beginning with classic or traditional controls such as proportional integral control (PI) [18], with methods of gains tuning [19], PID controls [2,4], PID with perturbation estimations [20], and adaptive gains with a feedforward approach [21]. However, linear control strategies are not the most suitable for nonlinear multivariate systems [22], and recent studies have focused on designing advanced controls such as MPC [23,24] and optimal controls [21]. The study of control techniques for biphasic separators is of research interest in academia and industry. Therefore, in this work, a virtualized biphasic separator is designed in an immersive environment, and as an application that shows its usefulness, different control techniques are proposed.

On the other hand, the use of virtual laboratories in educational applications requires that the students and professors have digital skills so that the teaching-learning process can be carried out. In [25], the authors show that the students with digital knowledge under study (future professors) do not have enough digital skills for the education of the new generations. In contrast, students who are young children show significant adaptation, enabling them to use digital technologies effectively. Thus, professors of the future must develop new technology skills.

In this research work, the design of an immersive virtual plant is proposed, for which the nonlinear and multivariable modeling of a biphasic separator is required, in which the correlation between the fluid level variable and the pressure of the gases when subjected to different hydrocarbon inputs is evident. One can observe an operation very similar to the real industrial process, even incorporating realistic monitoring interfaces, and thus providing a useful didactic tool in industrial and educational applications. A comparative analysis of three control strategies for a biphasic separator is proposed to test the usefulness of the virtual separator: a conventional one based on PID control and two model-based

strategies—a numerical controller and an MPC controller. The multivariable characteristics and the nonlinear nature of the process to be controlled in its modeling and virtualization are considered in this work.

The teaching-learning process is a transversal axis because the purpose of virtualizing a biphasic separator is usefulness in a virtual laboratory, which allows the application of the learning of mathematical models, as well as control strategies for the optimal performance of the virtual biphasic separator through the management of equipment, devices, and control variables to strengthen the knowledge of engineering students and professionals as well.

## 2. Description and Mathematical Modeling of the Biphasic Separator

### 2.1. Description of the Biphasic Separator

A biphasic separator is part of the oil collection process, located next to the production wells that extract the hydrocarbon by means of electro-submersible pumps. The product arrives from one or more wells through a set of valves and pipes called a manifold, providing data to account for production, record trends, and history.

Crude oil, so-called because it has not undergone any treatment, enters as a composition of phases (gas and fluid). Subsequently, these two components are separated within the container, as shown in the piping and instrumentation (P&ID) diagram in Figure 1. There is a gas outlet at the top of the container ($GF_{out}(t)$) and a fluid outlet at the bottom ($LF_{out}(t)$) [4].

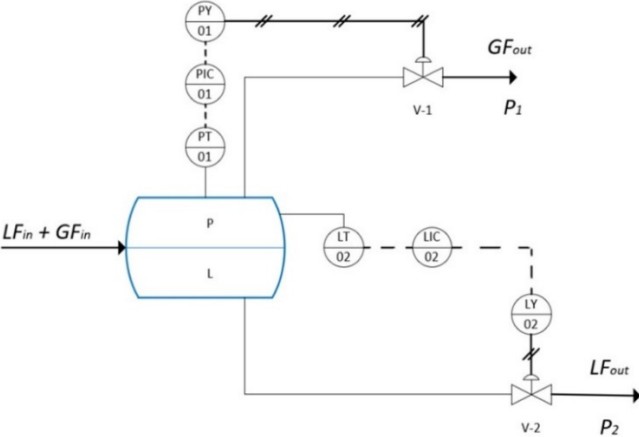

**Figure 1.** P&ID diagram of a biphasic separator.

The P&ID diagram shows the level control loop (02) with the respective level transmitter (LT), level indicator controller (LIC), level transducer (LY), and control valve for the level (V-2). Similarly, it shows the pressure control loop (01) with the respective pressure transmitter (PT), pressure indicator controller (PIC), pressure transducer (PY), and pressure control valve (V-1).

The separation operation is carried out due to several scenarios in the oil environment: (i) Considering the high corrosive and abrasion level of the gas, it is essential to isolate most of the fluid during its transportation to preserve the transport equipment and avoid pressure drops and capacity reductions in the transport lines. (ii) In a process in which the gas is flamed, usually this flow drags a high level of light oil, causing considerable losses, taking into account the fact that light oil has a high commercial value [4].

The biphasic separator, shown in Figure 1, has two inlet flows for liquid ($LF_{in}(t)$) and gas ($GF_{in}(t)$). These are directed to the horizontal tank. The fluid level and pressure variables in the tank are controlled using control valves for pressure (V-1) and control valves for levels (V-2), which are the actuators of the biphasic separator.

### 2.2. Mathematical Modeling of Biphasic Separation

Modeling is based on a mathematical description of the system's dynamic characteristics based on differential Equations that describe the system's dynamic behavior. The model is of a nonlinear multivariate type. In the modeling of separation systems on oil platforms, it is essential to mention that non-linearities exert a more significant impact on the dynamic behavior of these systems [23]. For mathematical modeling, the following conditions are considered: (i) the system in thermodynamic equilibrium, and (ii) the gas modeled as an ideal gas.

The nomenclature used in the mathematical modeling of the biphasic separator is described as follows: gas valve flow coefficient (CVG), liquid (water) valve flow coefficient (CVL), tank diameter ($D$) measured in meters, tank length ($C$) measured in meters, liquid level ($h_L(t)$) measured in meters, fluid input liquid ($LF_{in}(t)$) measured in m$^3$/s, fluid outflow liquid ($LF_{out}(t)$) measured in m$^3$/s, gas input flow ($GF_{in}(t)$) measured in m$^3$/s, gas outflow ($GF_{out}(t)$) measured in m$^3$/s, pressure in the tank ($P(t)$) measured in bars, downstream pressure of the gas valve (P1) measured in bars, downstream pressure of the liquid valve (P2) measured in bars, total tank volume ($V$) measured in m$^3$, volume of the liquid ($V_L(t)$) measured in m$^3$, gas volume ($V_G(t)$) measured in m$^3$, mass of gas in the tank ($M_G(t)$) measured in kg, liquid density ($\rho L$) measured in kg/ L, gas density ($\rho G(t)$) measured in kg/L, water density at 15.5 °C ($\rho$H$_2$O–15.5 °C) measured in kg/L, molar mass of the gas ($MMG$) measured in kg/mol, universal gas constant ($R$) measured in (bar × L)/(mol × K°), temperature ($T$) measured in K°, opening of the liquid valve ($aL(t)$), and opening of the gas valve ($aG(t)$).

The mathematical model of the level and pressure of the tank used considers the circular shape of the tank, adding the non-linearity condition to the model [23]. The total volume of the horizontal circular tank ($V$) is obtained using Equation (1). The dynamic of the liquid level of the biphasic separator tank is detailed in Equation (2).

$$V = \frac{\pi \times C \times D^2}{4}, \tag{1}$$

$$\frac{dh_L(t)}{dt} = \frac{LF_{in}(t) - LF_{out}(t)}{2C\sqrt{[D - h_L(t)]h_L(t)}}, \tag{2}$$

where the inlet flow of the liquid $LF_{in}(t)$ is a known value, whereas the outflow of the liquid $LF_{out}(t)$ depends on the control action applied to the valve of the fluid to $aL(t)$, as shown in Equation (3). The dynamic of the pressure of the separator tank is shown in Equation (4).

$$LF_{out}(t) = 2.4 \times 10^{-4} \times a_L(t) \times C_{VL} \times \sqrt{\frac{P(t) - P_2}{\frac{\rho_L}{\rho_{H_2O,15.5°C}}}}, \tag{3}$$

$$\frac{dP(t)}{dt} = \frac{P(t)[GF_{in}(t) - GF_{out}(t) + LF_{in}(t) - LF_{out}(t)]}{V - V_L(t)}, \tag{4}$$

where the inlet flow of gas $GF_{in}(t)$ is a known value, whereas the outflow of gas $GF_{out}(t)$ depends on the control action of the valve of the gas to $aG(t)$ and Equation (5).

$$GF_{out}(t) = 2.4 \times 10^{-4} \times a_G(t) \times C_{VG} \times \sqrt{\frac{(P(t) - P_1)(P(t) + P_1)}{\frac{\rho_G(t)}{\rho_{H_2O,15.5°C}}}} \tag{5}$$

Similarly, it is known that the density of the gas as a function of time is calculated by multiplying the pressure inside the tank by the molar mass of the gas, divided by the multiplication of the universal constant of gases and a specific temperature, as indicated

in Equation (6). It should be clarified that Equations (1)–(6) define the behavior of the biphasic separator.

$$\rho_G(t) = \frac{P(t)MM_G}{RT} \tag{6}$$

## 3. Methodology for Immersive Virtualization of a Biphasic Separator

The design methodology described in Figure 2 is used to develop the immersive virtualization of the biphasic separator, and the same can be applied to any industrial process virtualization. The first section corresponds to the mathematical modeling of the biphasic separator to be virtualized. The second stage corresponds to virtualization, in which the 3D modeling of the industrial process is included through the use of SolidWorks and 3Ds Max. Then, the plant dynamics are captured by the plant model, implemented in Unity. The third section designs the classic control algorithms and advanced controllers. Finally, it includes the interaction between the virtual plant and the controls using shared variables.

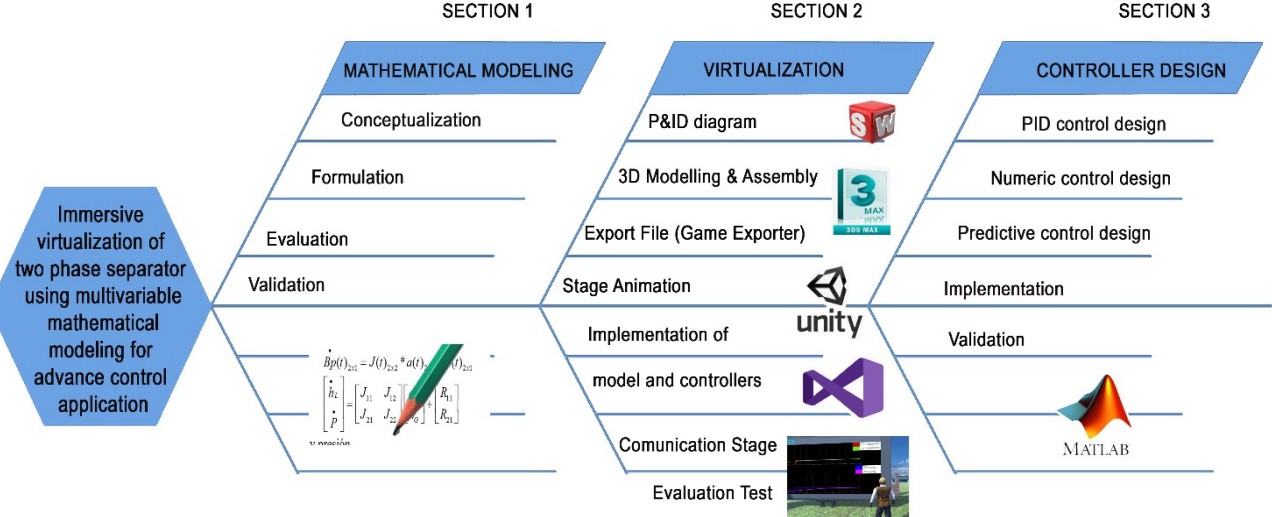

**Figure 2.** General diagram of the industrial process virtualization methodology.

### 3.1. Mathematical Modeling of the Industrial Process to Virtualize

The modeling is based on a mathematical description of the system's dynamic characteristics, in which a series of differential equations are used. The proposed model is of a multivariable type. To carry out the mathematical modeling of the virtualization process, the methodology described in Figure 2 is used, which begins with the conceptualization phase, in which several investigations are carried out to understand the physical phenomena present and to find the input and output variables that affect the industrial process. Then, in the formulation phase, Equations are proposed as a function of time that describe the nonlinear behavior of the process. In the evaluation phase, open loop tests are carried out with the Equations obtained, providing different values from the input variables to observe the output that represents the evolution of the system with respect to time. Finally, the validation is carried out, in which we analyze and verify that the results obtained are correct in order to terminate the mathematical modeling process; otherwise, it is necessary to return to performing revisions in the formulation phase. The mathematical model describing the behavior of a biphasic separator is described in Section 2.2.

### 3.2. Virtualization

The P&ID scheme shown in Figure 1 is the starting point. Then a computer-aided design (CAD) model of the industrial process is created through Autocad Plant 3D or Solidworks. Then the CAD model requires post-processing so that the models and animations are suitable for work in the graphics engine.

Unity 3D is a game engine developed by the Unity Company of Denmark, which is prominent due to its three-dimensional rendering capabilities, and is used to carry out both 2D and 3D projects. It has a simple and sufficiently powerful development environment, allowing the easy creation of video games and applications for various platforms and immersive environments [26]. Unity is the software that in this research work is used to create and operate control applications to provide immersive experiences, allowing solutions to be created in industrial areas, as well as training, simulation, and 3D experiences in these environments. This is accomplished thanks to the 3D environment design tools included in Unity through a visual editor and programming with the use of scripts

Unity includes animations, textures, sounds, and a variety of immersive components, which allow the user to interact with the virtualized scenario as it would occur in a real system. After designing the biphasic separator with its virtual components, the Unity graphics engine allows loading, rendering, and the adding of shadows and animations with their physical behavior. To achieve the dynamics of the change of the outputs against changes in the input, it is necessary to link the mathematical model with the virtualized physical elements. This is achieved by programming a script where the behavior of the variables is defined (mathematical model).

Mathematical models of industrial processes are encapsulated at this stage and thus can be integrated into controllers for process evaluation by including the control algorithm, and thereby obtaining synergy between the simulation and the controller. The mathematical model is programmed in Visual Studio (VS). VS is integrated into Unity; therefore, the mathematical model runs in Unity. This allows the user to monitor the dynamics of the process.

Shared memories are used to achieve communication between the control algorithms and Unity. First, the corresponding configurations are made in each software; after that, the communication is established, and the data exchange is validated in real time. Finally, the created memories are released.

We propose to use shared memory between the two software platforms for bilateral communication. This is an easy technique to apply, with short delays and low computational resource usage, avoiding the use of third-party functions. The shared memory method is part of Windows inter-process communications (IPC), providing advantages by linking processes using pre-allocated memory registers and no third-party features. The bidirectional data communication between Matlab and Unity, as shown in Figure 3, has been achieved with the need to call a dynamic link library (DLL), in which the shared memory method (SM) is implemented in the RAM. When the DLL manages the shared memory space, it also grants permissions for applications. For example, it grants permissions to modify/obtain the stored information and release the space when the application ends.

This process has the following phases: (i) The initial phase, in which the DLL can be instantiated using an identifier, in which the security and inheritance attributes, permissions to read/write reserved memory registers, RAM management, and labeling are declared. (ii) The execution phase, in which Unity 3D and Matlab must call a function to find the identifier through the tag and create a memory view, defining read/write permissions. The view allows one to update the records dedicated to each application. (iii) The shutdown phase, in which the shared memory is reserved while the process is running; when the application is closed, the memory must be freed by calling a function that ends with the reservation and labeling of the RAM so that it can be used by another system process [27].

Some scripts are required to link the control applications with the virtual plant. For example, in Unity, there is a script that describes the dynamic behavior of the plant. In addition, there are some scripts for visual effects, such as avatar manipulation, dynamic variable charts, instruments (valves and transmitters), among others. Moreover, the scripts related to the control algorithms (PID, numeric, and MPC) run in Matlab. Finally, each control algorithm is linked with the Unity application through communication scripts, allowing the exchange of data.

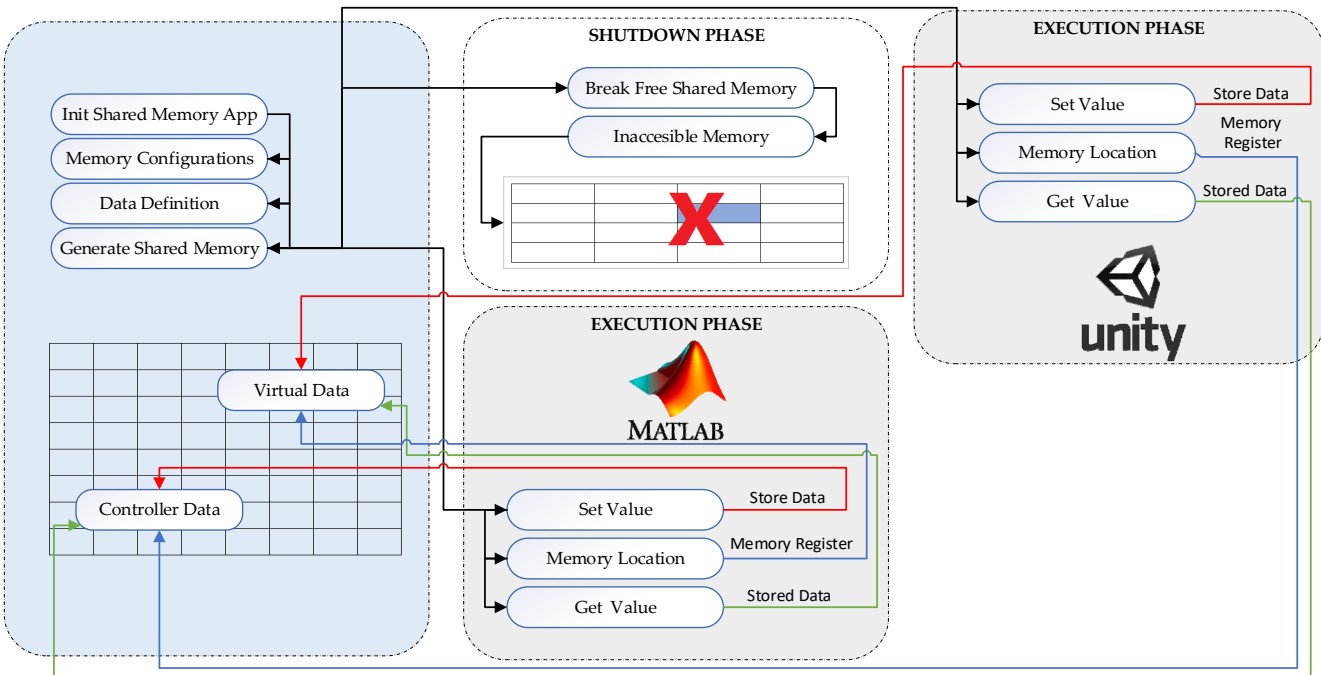

**Figure 3.** Communication between processes through shared memory.

### 3.3. Controller Design

The procedure described in Figure 2 is followed to develop the different control strategies. The design of control algorithms is first carried out; then, the designed controller is implemented and closed loop tests are carried out. Note that the controllers are designed on a different platform than the virtual plant, in which the biphasic separator operates. Finally, the performance of the controllers in response to changes in the reference and disturbances in the plant is validated and improved, a process for which the connection between the virtual platform and the controller described below is required. The control parameters of the evaluation are maximum overshoot, steady-state error, settling time, and actuator control action.

Once the virtual biphasic separator is available, we design, validate, and compare three control strategies: (i) PID, (ii) numerical control, and (iii) MPC, described in Section 4. Shared memories are used to achieve communication between the control algorithms and Unity. The corresponding configurations are made in each software platform, and then the communication is established and the data exchange is validated in real time. Finally, the created memories are released.

### 3.4. Mathematical Modeling of the Industrial Process to Virtualize

The teaching process applied to the students is shown in Figure 4. (i) In the prior knowledge stage, the user acquires theoretical knowledge regarding system modeling, system operation, and the design of the control strategies of the system. As a case study, the biphasic separator is used (pre-recorded video). In addition, the necessary files are provided, as well as the user manual of the virtual biphasic separator, where the minimum recommended technical requirements of the computer equipment are specified. Additionally, the users receive a laboratory guide developed by the professors with the activities to be carried out and a questionnaire to be filled out as the practice develops. Finally, the professor performs an induction for the use of the virtual separator (pre-recorded video) at this stage. (ii) In the interaction stage with the virtual biphasic separator, the practice is executed. First, the users interact with the virtual environment, then design and evaluate the controllers in the virtual plant. (iii) In the evaluation stage, the users are subjected to a questionnaire. (iv) Finally, feedback is provided by the course professor to the students.

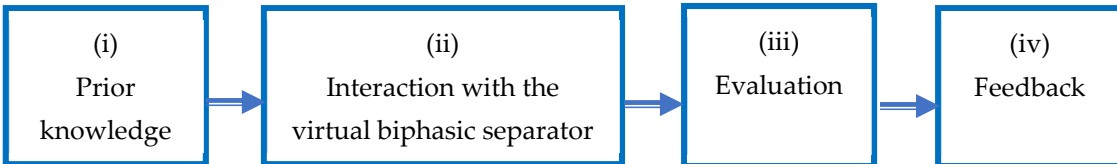

**Figure 4.** The procedure used in the teaching process.

## 4. Application Design of Virtual Biphasic Separator Control Strategies

The study of control strategies is fundamental for managing the variables involved in closed-loop systems. In control theory, a control strategy rules the dynamic behavior of a system (process) by regulating an output variable to follow a reference (setpoint) by means of an input variable. There are classical control strategies, such as PI controllers, and more advanced control strategies, such as numerical and MPC controllers. Advanced control strategies have the advantages of controlling multiple input–multiple output processes and show better performance as they are based on a model of the process being controlled. The majority of processes in the industrial sector use one of the previously mentioned control strategies to work autonomously without constant supervision by human personnel.

This section presents the designs of three control strategies: PID control, numerical control, and MPC control, aiming to determine the strategy with the best performance.

### 4.1. Design of a PID Control Algorithm for a Biphasic Separator

The PID controller is the most common control algorithm. Most feedback loops are controlled using this algorithm or another with slight variations. The PID algorithm can be described as indicated in Equation (7):

$$u(t) = \left( Ke(t) + \frac{K}{T_i} \int_0^t e(t)dt + KT_d \frac{de(t)}{dt} \right) \tag{7}$$

where $u(t)$ is the control signal, $e(t)$ is the error, $\int_0^t e(t)dt$ is the integral of the error, and $de(t)/dt$ is the error derivative. The controller parameters are proportional gain $K$, integral gain $K_i = K/T_i$ where $T_i$ is integral time, and derivative gain $K_d = KT_d$ where is derivative time $T_d$ [28,29].

Two PID control loops are implemented in the biphasic separator, one for each variable, as indicated in the closed loop diagram in Figure 5.

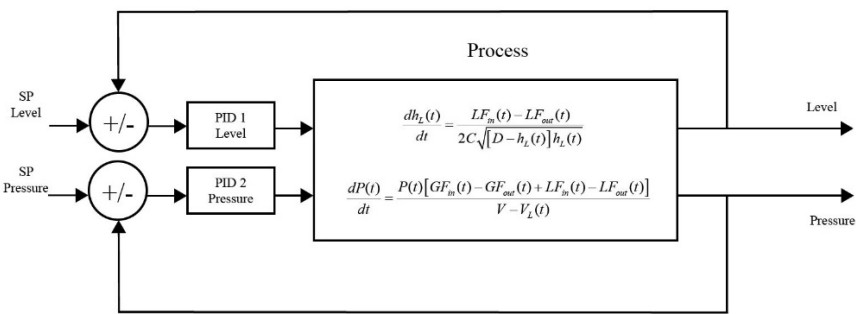

**Figure 5.** PID control loop implementation diagram.

The constants of each control loop for level and pressure are tuned with the Lambda tuning technique [15], and after a fine-tuning process, the values are indicated as follows: for the level controller ($Kp = 3$, $K_i = 0.03$ and $K_d = 0$), for pressure controller ($Kp = 0.36$, $K_i = 0.01$ and $K_d = 0$).

### 4.2. Design of a Numerical Control Algorithm for a Biphasic Separator

The numerical controller is a controller based on the plant model; therefore, the mathematical model is needed, in addition to the knowledge of some additional topics detailed below, such as Euler's method and Markov chains that facilitate the writing of the separator model in a matrix form. Euler's method predicts the next value of a function using the definition of the derivative [30].

Markov chains are used for the quantification of future errors, since this process is a random phenomenon, dependent on time, which means that the future depends only on the present and not on the past. In other words, the past and the future are independent, as long as the present is known [31].

It is necessary to express the equations that describe the behavior of the process as a function of matrices and vectors to perform the design of the numerical controller, which is detailed in the following steps:

(i)　Equation (3) is substituted into Equation (2)—with both Equations having been obtained during the mathematical modeling of the biphasic separator—to obtain an Equation representing the plant's dynamics as a function of the parameters that vary in time, as shown in Equation (8).

$$\frac{dh_L(t)}{dt} = -\frac{2.4 \times 10^{-4} \times C_{VL} \times \sqrt{\frac{P(t)-P_2}{\frac{\rho_L}{\rho_{H_2O,15.5°C}}}}}{2C\sqrt{[D - h_L(t)]h_L(t)}} a_L(t) + \frac{LF_{in}(t)}{2C\sqrt{[D - h_L(t)]h_L(t)}} \tag{8}$$

(ii)　The terms of Equation (8) are designed as elements of the matrix J (Jacobian) and the vector R (process behavior), obtaining the element $J_{11}$ represented in Equation (9), the element $J_{12}$ represented in Equation (10), and the element $R_{11}$ represented in Equation (11).

$$J_{11} = -\frac{2.4 \times 10^{-4} \times C_{VL} \times \sqrt{\frac{P(t)-P_2}{\frac{\rho_L}{\rho_{H_2O,15.5°C}}}}}{2C\sqrt{[D - h_L(t)]h_L(t)}} \tag{9}$$

$$J_{12} = 0 \tag{10}$$

$$R_{11} = \frac{LF_{in}(t)}{2C\sqrt{[D - h_L(t)]h_L(t)}} \tag{11}$$

(iii)　Similarly, Equation (5) is used in Equation (4)—with both Equations having been obtained in Section 2.2—and the dynamic of pressure is expressed as a function of the variables of level and pressure, as shown in Equation (12).

$$\frac{dP(t)}{dt} = -\frac{P(t) \times 2.4 \times 10^{-4} \times C_{VL} \times \sqrt{\frac{P(t)-P_2}{\frac{\rho_L}{\rho_{H_2O,15.5°C}}}}}{V - V_L(t)} a_L(t) - \frac{P(t) \times 2.4 \times 10^{-4} \times C_{VG} \times \sqrt{\frac{(P(t)-P_1)(P(t)+P_1)}{\frac{\rho_G(t)}{\rho_{H_2O,15.5°C}}}}}{V - V_L(t)} a_G(t) + \frac{P(t)[GF_{in}(t) + LF_{in}(t)]}{V - V_L(t)} \tag{12}$$

(iv)　Then, the terms of Equation (12) are assigned as elements of the matrix J (Jacobian) and the vector R (process behavior), which is detailed below in the section regarding the design of the numerical controller. The element $J_{21}$ represented in Equation (13), the element $J_{21}$ represented in Equation (14), and the element $R_{21}$ represented in Equation (15) are obtained.

$$J_{21} = -\frac{P(t) \times 2.4 \times 10^{-4} \times C_{VL} \times \sqrt{\frac{P(t)-P_2}{\frac{\rho_L}{\rho_{H_2O,15.5°C}}}}}{V - V_L(t)} \tag{13}$$

$$J_{22} = -\frac{P(t) \times 2.410^{-4} \times C_{VG}\sqrt{\frac{(P(t)-P_1)(P(t)+P_1)}{\frac{\rho_G(t)}{\rho_{H_2O,15.5°C}}}}}{V - V_L(t)} \tag{14}$$

$$R_{21} = \frac{P(t)[GF_{in}(t) + LF_{in}(t)]}{V - V_L(t)} \tag{15}$$

(v)   Next, to express the multivariate mathematical model of the biphasic separator in the form of a matrix, the structure shown in Equation (16) is used

$$\dot{B}(t)_{2x1} = J(t)_{2x2} \times a(t)_{2x1} + R(t)_{2x1} \tag{16}$$

where $\dot{B}(t)$ is the vector of the derivatives of level and pressure, $J(t)$ is the Jacobian Matrix, $a(t)$ is the vector of opening valves, and $R(t)$ is the vector of the behavior of the plant.

(vi)  Finally, using the elements of the matrix J and the vector R detailed in Equations (9)–(11), and (13)–(15) and using as a basis the structure of Equation (16), the model is represented in the form of a matrix in Equation (17), which will be used to propose the design of the numerical controller.

$$\begin{bmatrix} \dot{h_L} \\ \dot{P} \end{bmatrix} = \begin{bmatrix} J_{11} & J_{12} \\ J_{21} & J_{22} \end{bmatrix} \begin{bmatrix} a_L \\ a_G \end{bmatrix} + \begin{bmatrix} R_{11} \\ R_{21} \end{bmatrix} \tag{17}$$

For the design of the controller, it is necessary to apply Euler's method to Equation (16) to obtain the discrete model of the process indicated in Equation (18), where $B(k)$ is the vector of the level and pressure values in sample k, and $T_o$ is the sampling time. This discrete model predicts the following values $B(k+1)$ shown in Equation (19), which is obtained after performing mathematical operations.

$$\frac{B(k+1) - B(k)}{T_o} = J(k)a(k) + R(k) \tag{18}$$

$$B(k+1) = T_o[J(k)a(k) + R(k)] + B(k) \tag{19}$$

Next, Markov chains are applied to the term $B(k+1)$ of Equation (19), where $Bd(k+1)$ expresses the desired values of level and pressure. $W$ is a square $2 \times 2$ matrix of weights used to decrease the error as detailed in Equation (20).

$$B(k+1) = Bd(k+1) - W[Bd(k) - B(k)] \tag{20}$$

Equations (19) and (20) are equalized, obtaining Equation (21). The law of control in Equation (22) is shown, which will be implemented to perform the simulation of the controller.

$$Bd(k+1) - W[Bd(k) - B(k)] = T_o[J(k)a(k) + R(k)] + B(k) \tag{21}$$

$$a(k) = J^{-1}\left(\frac{\{Bd(k+1) - W[Bd(k) - B(k)] - B(k)\}}{T_o} - R(k)\right) \tag{22}$$

To find the tuning gains of the numerical controller, it is necessary to obtain the matrix $W$, for which Equation (18) is used, to obtain control actions of the valves $a(k)$, as observed in Equation (23).

$$a(k) = J^{-1}(k)\left(\frac{B(k+1) - B(k)}{T_o} - R(k)\right) \tag{23}$$

Equalizing Equations (22) and (23), Equation (24) is obtained, which after performing mathematical operations forms Equation (25).

$$J^{-1}(k)\left(\frac{B(k+1)-B(k)}{T_o}-R(k)\right)=J^{-1}(k)\left(\frac{\{Bd(k+1)-W[Bd(k)-B(k)]-B(k)\}}{T_o}-R(k)\right) \qquad (24)$$

$$Bd(k+1)-B(k+1)=W[Bd(k)-B(k)]$$

$$e_B(k+1)=W[e_B(k)] \qquad (25)$$

The error vector $e_B(k)$ of dimension $1\times 2$ comes from the difference between the desired values $Bd(k)$ and the current values $B(k)$, whereas the error vector of $e_B(k+1)$, of dimension $1\times 2$, comes from the difference between the desired values $Bd(k+1)$ and current values $B(k+1)$. Next, Equation (25) of the vector of errors $e_B(k)$ of dimension $1\times 2$ is extended, which has as elements the error level $e_{h_L}(k)$ in the position $e_B(k)_{1,1}$ and the pressure error $e_P(k)$ in the position $e_B(k)_{2,1}$, as observed in Equation (26).

$$e_B(k)=\begin{bmatrix} e_{h_L}(k) \\ e_P(k) \end{bmatrix} \qquad (26)$$

In the weight matrix $W$, the value of the level weights is the element $W_{1,1}$, and the pressure weight is the element $W_{2,2}$, indicated in Equation (27).

$$W=\begin{bmatrix} W_{1,1} & 0 \\ 0 & W_{2,2} \end{bmatrix} \qquad (27)$$

To find the values of the tuning weights of the numerical controller belonging to the matrix W, we proceed to the process detailed as follows:

(a) Using Equation (24) to represent the error of level $e_{h_L}(k)$ in Equation (28), using the elements of the first row and the first column of the vector $e_B(k)$ of Equation (26) and the weight matrix $W$ of Equation (27),

$$e_{h_L}(k+1)=W_{1,1}\left[e_{h_L}(k)\right]. \qquad (28)$$

(b) Equation (28) is evaluated by assigning values to the sample "$k$" to obtain the evolution of the level errors over time, as shown in Equations (29)–(31).

$$\text{With } k=1 \qquad e_{h_L}(2)=W_{1,1}\left[e_{h_L}(1)\right] \qquad (29)$$

$$\text{With } k=2 \qquad e_{h_L}(3)=W_{1,1}\left[e_{h_L}(2)\right]=W_{1,1}^2\left[e_{h_L}(1)\right] \qquad (30)$$

$$\text{With } k=\text{n} \qquad e_{h_L}(n)=W_{1,1}^{(n-1)}\left[e_{h_L}(1)\right] \qquad (31)$$

(c) After analyzing Equations (29)–(31), it is determined that the term $W_{1,1}^{(n-1)}$ of Equation (31) has a similarity with the exponential function detailed in Equation (32), so the term $W_{1,1}^{(n-1)}$ is equalized with the exponential function, obtaining the result indicated in Equation (33).

$$f(x)=a^x \qquad (32)$$

$$a^x \cong W_{1,1}^{(n-1)} \qquad (33)$$

(d) We proceed to assign different values to $W_{1,1}^{(n-1)}$ of Equation (33) to determine the gain of the level process, and the range of values is established in Equation (34).

$$0<W_{1,1}<1 \qquad (34)$$

(e) To obtain the gain of the pressure process, the four steps mentioned above must be repeated, but with the respective data of the error vector and the gain matrix, that is,

the pressure error $e_P(k)$ and the pressure weight $W_{2,2}$, obtaining the interval defined by Equation (35):

$$0 < W_{2,2} < 1 \tag{35}$$

(f) The gains of the tuning weights of the numerical controller of the matrix $W$ are obtained by trial and error, respecting the restrictions of the intervals indicated in Equations (34) and (35). The values detailed are the following: level weight ($W_{1,1}$ = 0.99) and pressure weight ($W_{2,2}$ = 0.99). The numerical control algorithm is shown in Figure 6.

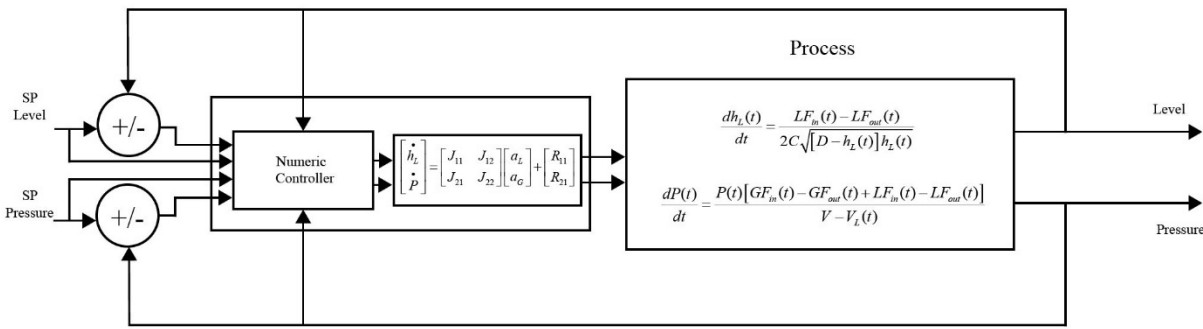

**Figure 6.** Diagram of implementation of numerical control loops.

### 4.3. Design of a Predictive Control Method Based on a Model MPC for a Biphasic Separator

Predictive control uses a process model to obtain the control signal while minimizing an objective function. The same structure has the following elements [31]: (i) the explicit use of a model to predict the evolution of the process in future moments; (ii) the minimization of a cost function; (iii) the use of a finite and sliding control horizon that involves the calculation of the control sequence for the entire horizon, but with the application of the first signal of the sequence and the repetition of the whole process in the next sampling time.

A model is used to predict the evolution of the output or process state from the known input and output signals. Then, future control actions are calculated with an optimizer, which considers the objective function and possible constraints [32].

Equation (36) details the objective function (OF), which uses the mathematical model of the process to calculate the control action's predictions. The OF minimizes the steady-state error between variables to be controlled and references, while minimizing changes in control action.

$$J(N_1, N_2, N_u) = \sum_{j=N_1}^{N_2} \delta(j)[\hat{y}(t+j|t) - w(t+j)]^2 + \sum_{j=N_1}^{N_u} \lambda(j)[\Delta u(t+j-1)]^2 \tag{36}$$

where $N_1$ and $N_2$ are the minimum and maximum prediction horizon, whereas $N_u$ is the control horizon, $\delta(j)$ and $\lambda(j)$ are weighting sequences, $\hat{y}(t+j|t)$ is the optimal prediction of the output, $j$ steps forward calculated with known data at the time instant $t$,$w(t+j)$ is the future reference trajectory, and $\Delta u(t+j-1)$ is the control action.

The objective function for MIMO systems, detailed in Equation (37), seeks to minimize the error of level and error of pressure. At the same time, the variations of the control actions of the actuators, i.e., the liquid valve and the gas valve.

$$J(k) = \sum_{u1,u2}^{N_p} \sum_{i=N_w} \delta_1(k)\left[\hat{h}(k+i|k) - hd(k+i|k)\right]^2 + \delta_2(k)[\hat{p}(k+i|k) - pd(k+i|k)]^2 + \sum_{i=0}^{N_c-1} \lambda_1(k)$$
$$[\Delta u_1(k+i-1)]^2 + \lambda_2(k)[\Delta u_2(k+i-1)]^2 \tag{37}$$

Subject to:

$$\begin{aligned}
\Delta u_{\min} &\leq \Delta u_1 \leq \Delta u_{\max} \\
\Delta u_{\min} &\leq \Delta u_2 \leq \Delta u_{\max}
\end{aligned} \tag{38}$$

$$\hat{h}_{\min} \leq \hat{h} \leq \hat{h}_{\max} \tag{39}$$

$$\hat{p}_{\min} \leq \hat{p} \leq \hat{p}_{\max} \tag{40}$$

In Equation (37), $N_w$ and $N_p$ are the beginning of the prediction horizon and the number of samples of the prediction horizon, respectively, whereas $N_c$ is the control horizon itself, which should always be less than the prediction horizon. $\hat{h}(k+i|k)$ is the predicted level output and $\hat{p}(k+i|k)$ is the predicted pressure output, whereas $hd(k+i|k)$ is the desired level value and $pd(k+i|k)$ is the desired pressure value.

The operating conditions of the processes in the optimization problem are included as inequality constraints. The first constraint of the optimization problem, indicated in Equation (38), is the percentage of the fluid valve opening $\Delta u_1$, which is responsible for manipulating the level and the percentage of the opening of the gas valve $\Delta u_2$ accountable for managing the pressure of the tank. Both have the maximum value $\Delta u_{\max} = 1$ and the minimum value $\Delta u_{\min} = 0$.

The second inequality constraint of Equation (39) describes the tank level limits, which are $h_{\min} = 0 [\mathrm{m}]$ and $h_{\max} = 3 [\mathrm{m}]$. Finally, the third inequality constraint of Equation (40) represents the pressure limits of the tank $p_{\min} = 7 [\mathrm{bar}]$ and $p_{\max} = 50 [\mathrm{bar}]$.

The pressure and the level share the same number of N samples in the prediction and control horizons $N_c$. The values of the weighting parameters for predictive control are determined by means of the trial and error method, in which the prediction horizon takes 18 samples $N_w = 18$, and the control horizon takes eight samples $N_c = 8$ every 0.1 s, i.e., it is analyzed in a time horizon equivalent to $18 \times 0.1$ s. It is important to note that the prediction horizon is greater than the control horizon.

The values of the parameters $\delta_1, \delta_2$ correspond to the weight of the error, and the values of the constants $\lambda_1, \lambda_2$ correspond to the weight assigned to the variations of the control valves. The parameters are determined by means of the trial and error method, and the parameters are following: level prediction horizon $N_w = 18$, pressure prediction horizon $N_w = 18$, level horizon control $N_c = 8$, pressure horizon control $N_c = 8$, weight related to the level error $\delta_1 = 100$, weight related to the level pressure error $\delta_2 = 0.1$, weight related to the variation of level control actions $\lambda_1 = 0.001$, weight related to the variation of pressure control actions $\lambda_2 = 0.001$.

Based on Equations (2) and (4) described in Section 2.2, the prediction model implemented in the controller is obtained. Equation (41) describes the prediction of the level behavior $\hat{h}$, and Equation (42) represents the prediction of pressure behavior $\hat{P}$.

$$\hat{h}(k+i|k) = \frac{L_{in}(k) - L_{out}(k)}{2C\sqrt{[D - h(k)]h(k)}} \tag{41}$$

$$\hat{P}(k+i|k) = \frac{P(k)[G_{in}(k) - G_{out}(k) + L_{in}(k) - L_{out}(k)]}{V - V_L(k)} \tag{42}$$

The MPC control algorithm of level and pressure are implemented as shown in Figure 7.

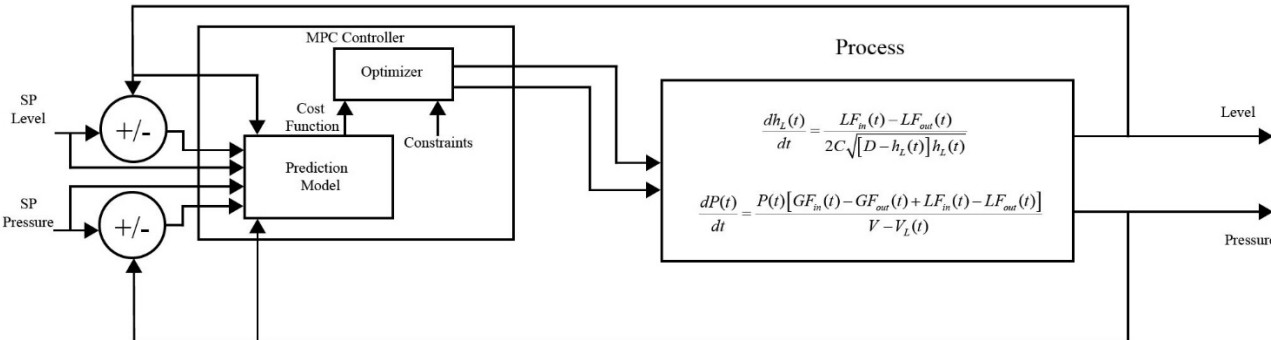

**Figure 7.** MPC control loop implementation diagram.

## 5. Results

### 5.1. Virtual Biphasic Separator System Performance

The parameters used to describe the virtualized separator, such as tank dimensions and other process constants, were as follows: $C = 8$ m; $D = 3$ m; $C_{VG} = 120$; $C_{VL} = 1025$; $P1 = 6$ bar; $P2 = 6$ bar; $\rho L = 850$ kg/L; $\rho H_2O_{(15.5\,°C)} = 99.2$ kg/L; $MM_G = 0.029$ kg/mol; $R = 0.08314474$ (bar $\times$ L)/(mol $\times$ K°); $T = 303.15$ K°; $V = 56.6$ m$^3$.

The virtual environment in an immersive environment of the biphasic separator is shown in Figure 8. It can be observed that the environment is similar to that of a real-life plant, which includes real instrumentation, monitoring screens, and the dynamics of the nonlinear and multivariable process with enhanced realism.

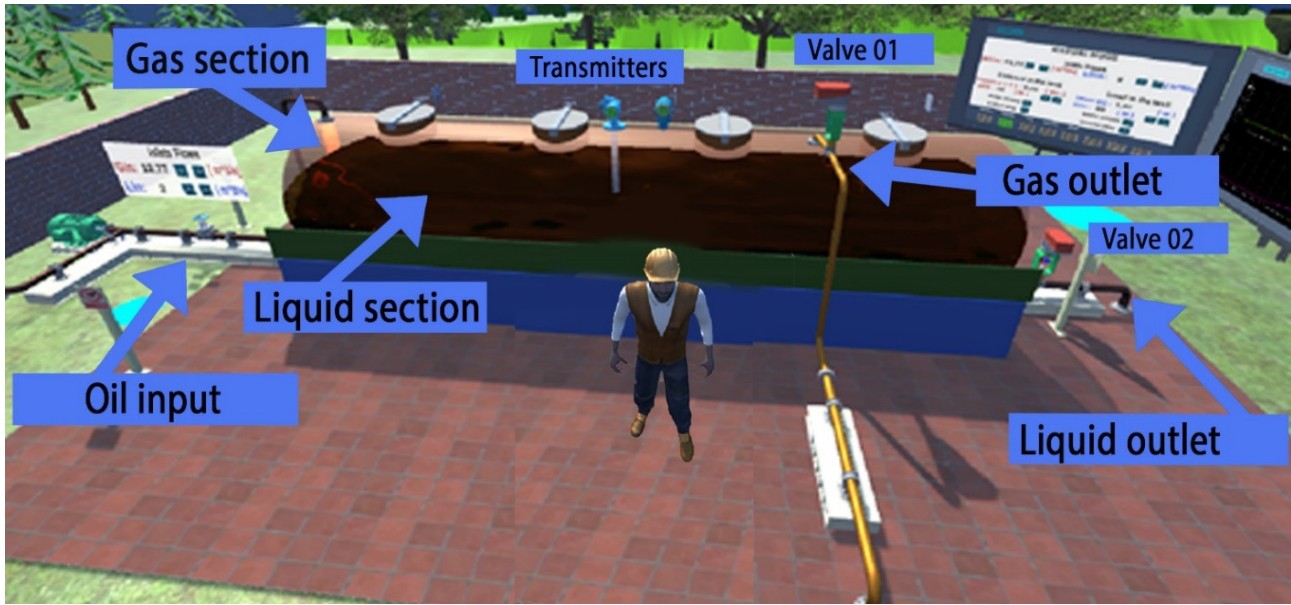

**Figure 8.** Virtualized biphasic separator.

The description and driver screen interaction are operated by using the function keys, as shown in Figure 9.

In this way, following the proposed methodology, the mathematical modeling of the biphasic separator and the virtualization of the industrial environment, containing the biphasic separator, display and control screens, control room, instrumentation components, and animations of its elements, have been developed. The result is an immersive industrial environment that is quite similar to reality.

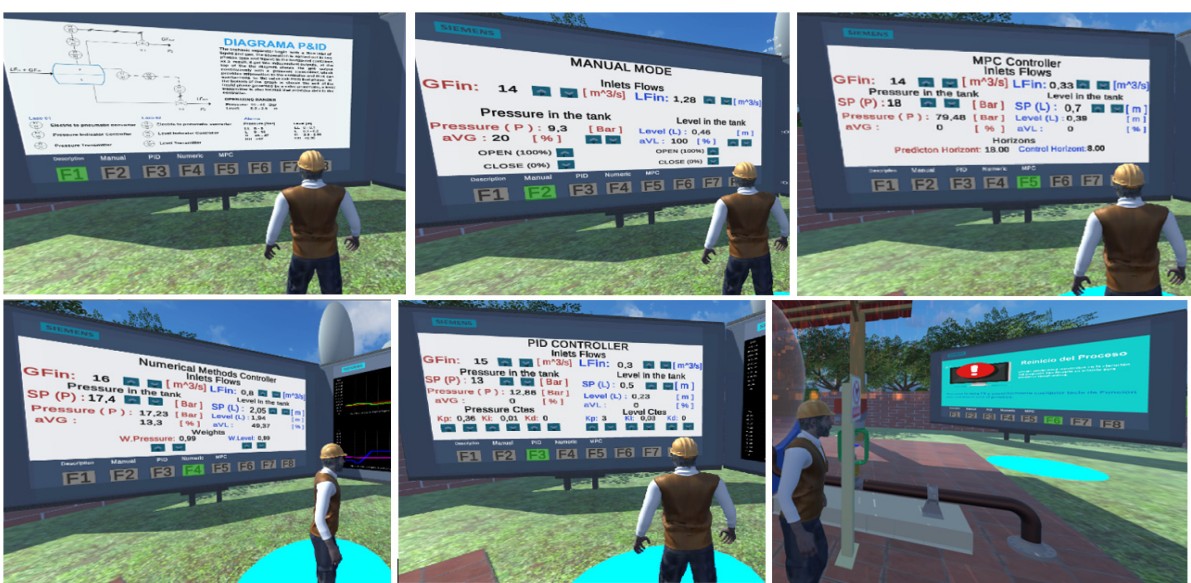

**Figure 9.** Interaction with screens and function keys.

A questionnaire regarding usability was created in an electronic multiple-choice form. The questionnaire contained fourteen questions. The sample addressed consisted of 48 students of engineering and two automation engineers. The first two questions provided basic information about the activities (Figure 10a) and previous knowledge of the use of virtual environments (Figure 10b). As can be seen, 88% of the survey respondents had not used a virtual learning application before.

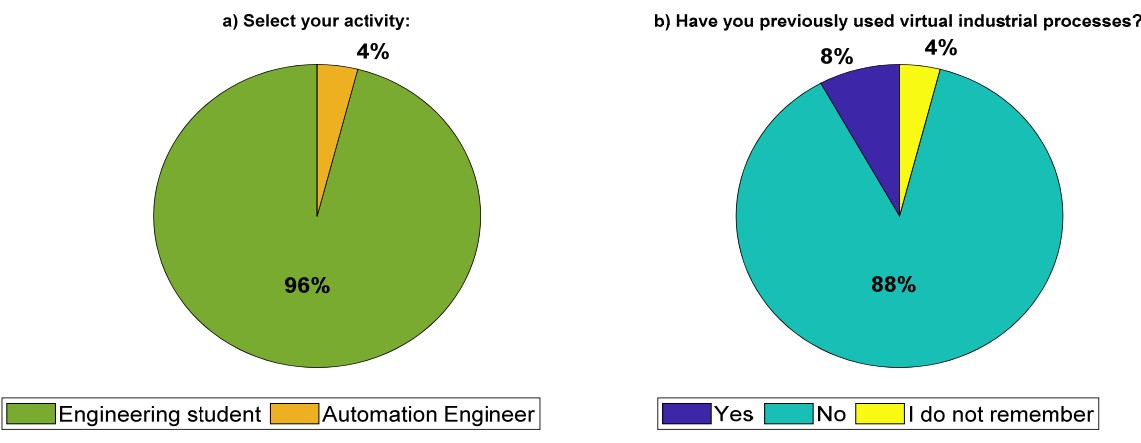

**Figure 10.** Questions regarding general information. (**a**) First question, regarding activities. (**b**) Second question, regarding previously used virtual reality applications.

The next section of questions evaluated the performance of the virtual biphasic separator; for instance, Figure 11 shows the answer to questions 3 and 5, showing that most of the survey respondents said that the application was intuitive (86%), and noted a positive level of realism (63% very high and 32% high).

The other group of questions evaluated the degree of acceptance of working with these learning tools; for instance, Figure 12 shows the answers to questions 10 and 11. The results show that 75.5% showed a very high acceptance level, and 16.3% showed a high level of acceptance (Figure 12a). Furthermore, most of those surveyed were interested in learning in virtual laboratories (87%) (Figure 12b). Figure 13 shows that 79.6% noted a positive impact on the learning procedure. The authors of [33] showed that an application has good performance if the usability is higher than 75%. In this proposal, the results show that the usability and acceptance of this application were higher than 75%. In this

context, the virtual learning tool allows the training of students or engineers in process and automatic control.

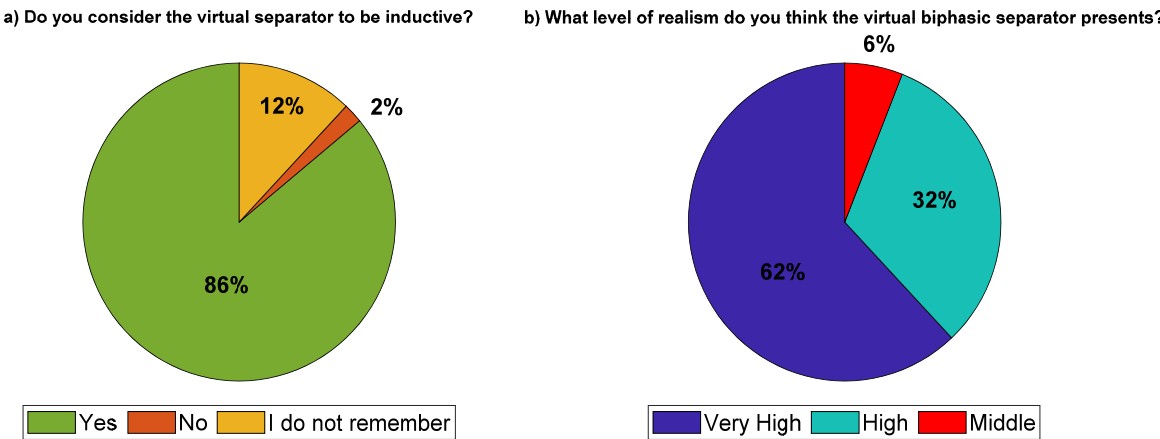

**Figure 11.** Questions regarding the performance of the virtual biphasic separator. (**a**) Third question, regarding the use of the application. (**b**) Fifth question, regarding realism.

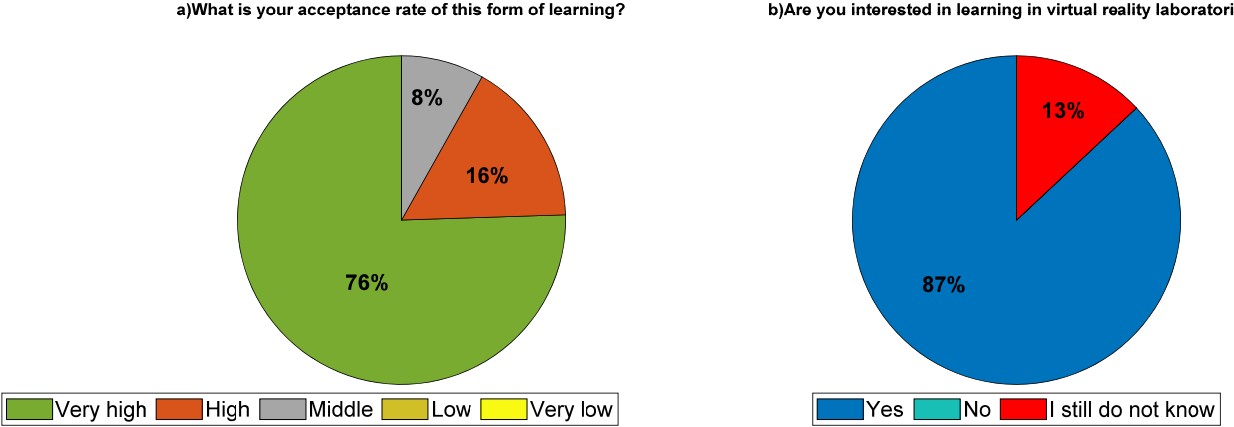

**Figure 12.** Questions regarding the degree of user satisfaction. (**a**) Tenth question, regarding the acceptance rate. (**b**) Eleventh question, regarding the interest of use.

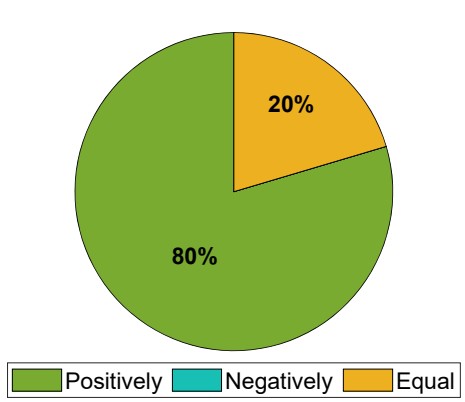

**Figure 13.** Results of question 14, regarding changes in learning.

Finally, 20 students of the control systems course answered a quiz to evaluate their knowledge regarding controller design. The maximum score was 10 points. The students achieved an average of 7.1, and they did not use the virtual learning tool in their lectures. In the second course of 19 students who answered the same quiz, they obtained an average

score of 8.9, and in their lectures they used the virtual learning tool. Based on the quiz results, there was an increase in the score by 1.8 points, equivalent to 17%. We can thus conclude that the proposal enables an improvement in the teaching-learning process.

### 5.2. Performance of the Control Strategies of a Biphasic Separator

In this section, we analyze the control strategies applied to the virtual reality biphasic separator. It is important to mention that before implementing the control strategies, the students had studied mathematical modeling, the design of control techniques, and biphasic separator behavior in theoretical classes.

The variables of level and pressure were controlled, reaching their respective set points, whereas the flows that were supplied (inlets) presented changes in the gas inlet flow, although the fluid flow was kept constant, as follows: (i) Fixed fluid inlet supply of 0.2 m$^3$/s (LFIN); (ii) a variable gas inlet flow (GFIN) supply starting with a value of 10 m$^3$/s, then at 900 s, the value rose to 15 m$^3$/s, and finally, at 1400 s, the value dropped to 3 m$^3$/s.

**Analysis of the Variable Level**

Figure 14 shows the response of the front level (orange), where one can see the evolution of the PID controller (green), the numerical controller (magenta), and the MPC controller (blue). It is important to mention that the change of gas input flow is a disturbance for the level variable.

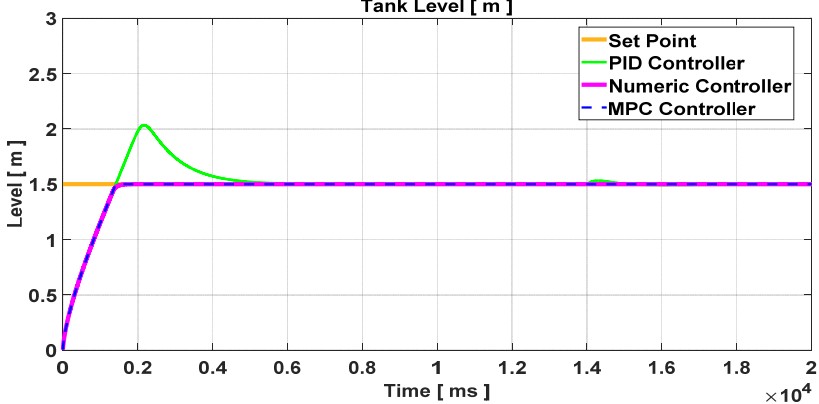

**Figure 14.** Performance of control algorithms in the variable level.

The level setpoint value was kept constant at 1.5 m at all times (Figure 14). Furthermore, it was observed that the response of the PID controller presented an overshoot at 200 s and also had a high settling time. On the other hand, the numeric and MPC controllers did not present an overshoot and were even faster than the PID controller. Finally, at 1400 s, a decrease in the gas flow input disturbed the variable level, which most noticeably affected the PID controller, causing an increase in the level, which was immediately stabilized, whereas the numeric and MPC controllers were not affected by the disturbance.

Table 1 compares the three controllers in terms of the performance parameters of overshoot, steady-state error, and settling time. The results enable us to conclude that the MPC controller shows better performance. The latter showed the best results under the operating conditions evaluated: the single-level setpoint and disturbance in the gas flow input. In addition to the controller's performance, it is also essential to consider the performance of the actuator control actions, as it must be verified that they are not abrupt, in order to protect the lifespan of the actuators.

**Table 1.** Comparison of PID, numeric, and MPC controllers in terms of the level variable.

| Parameters | PID | Numerical | MPC |
|---|---|---|---|
| Overshoot (%) | 17.6 | 0 | 0 |
| Settling time (s) | 423.4 | 80.6 | 137.8 |
| Steady-state error (m) | 0 | 0 | 0 |

**Analysis of the Pressure Variable**

Figure 15 shows the pressure response. Again, the setpoint is depicted in orange, the evolution of the variable with a PID controller is in green, the numeric controller is in magenta, and the MPC controller is in blue.

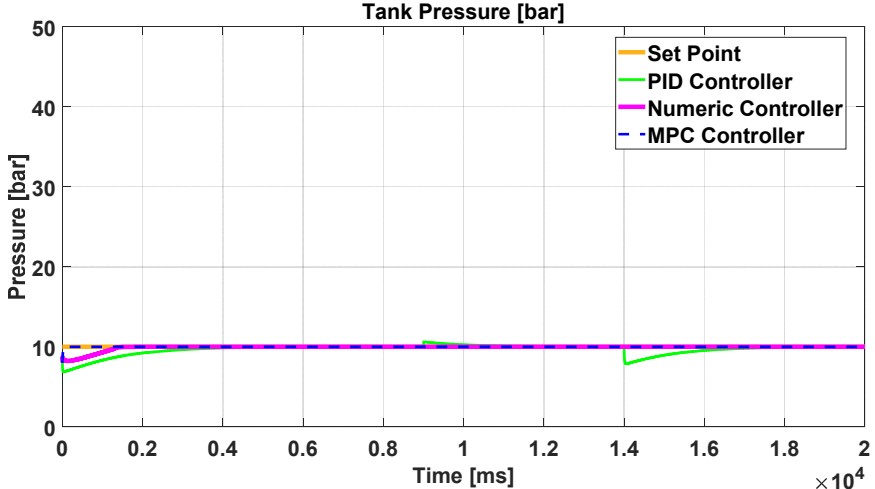

**Figure 15.** Performance of control algorithms in relation to the pressure variable.

The simulation kept the pressure setpoint value constant at 10 bar at all times. In the first seconds of the simulation, the response of the three controllers did not present overshoots; however, the PID controller had a settling time of 252 s. At 900 s, there was an increase in the input gas flow, but this only caused a small load on the process. The PID control showed an increase in pressure, whereas the numeric and MPC controllers did not present changes in the pressure value. Finally, at 1400 s, there was a decrease in the gas flow ingress, exerting a load on the process. This most noticeably affected the PID controller, causing a decrease in pressure, but it stabilized at 1600 s, whereas the numeric and MPC controllers were not affected by the disturbance.

Table 2 compares the three controllers in terms of the performance parameters of overshoot, steady-state error, and settling time. The results show that the performance of the MPC controller was the most outstanding because it had the best settling time under the evaluated operating conditions. There was a single setpoint of the pressure and disturbances in the gas flow input. In addition to the controller's performance, it is also essential to consider the performance of the actuator control actions, since it must be verified that they are not abrupt, in order to protect the lifespan of the actuators.

**Table 2.** Comparison of PID, numerical, and MPC performance in terms of the pressure variable.

| Parameters | PID | Numerical | MPC |
|---|---|---|---|
| Overshoot (%) | 0 | 0 | 0 |
| Settling time (s) | 252.9 | 11.2 | 1.8 |
| Steady-state error (bar) | 0 | 0 | 0 |

## 6. Conclusions

The proposed methodology of process virtualization allows one to obtain a realistic immersive virtual environment for industrial processes, which considers the multivariable dynamics with the correlations between the variables to control the fluid liquid and gas flow. The virtual plant allowed the evaluation of various control algorithms for a biphasic separator and allowed the determination of its optimal performance for training students or engineers in process and automatic control. The performance tests demonstrated its high flexibility in validating different control techniques. The results show that the proposed virtual reality biphasic separator is intuitive and easy to use. Moreover, the virtual plant improves the learning process. The average score obtained by the students in their final exam showed an increase of 17% when the virtual plant was used. Thus, this learning tool is currently used to teach courses related to control techniques applied to industrial processes.

Three control strategies for the biphasic separator were designed, evaluated, and compared. We first analyzed the use of a traditional PID controller for liquid level and gas pressure. Then, a numerical controller and a model-based predictive controller (MPC) were designed. The numeric and MPC controllers showed better performance in the transient and steady state of the controlled variables than the PID controller.

Steady-state errors were approximately equal to zero in all control strategies. However, in the transient state, the numerical and MPC controllers presented better performance when the maximum overshoot and settling time were evaluated.

Tests showed that the PID controller allows one to control the two variables separately and to obtain errors around zero in a steady state. However, the PID controller presents worse performance in the transient state than numerical and MPC controllers in terms of maximum overshoot and settling time.

On the other hand, the control actions were analyzed to determine the activation of the actuators. The numerical and MPC controller displayed smooth control actions that would extend the actuator's lifespan, compared to the PID controller.

Since MPC and the numerical controller are based on the plant model, they have a better performance than traditional PID controllers in terms of setpoint changes of the liquid level and gas flow and disturbances produced by molecular changes in the input hydrocarbons.

The MPC control strategy designed for the biphasic separator, unlike the numerical controller, presents a faster reaction to changes in operating points, with a settling time that is 9.4 s less than that of the numerical controller in the pressure variable. This is because, as part of the design, it considers prediction models of the variables of interest of the process, allowing the controller to anticipate events.

In addition, the MPC controller allows one to improve or worsen the operation of actuators by tuning the control algorithm weights. This feature is not available in the other designed controllers.

The virtual reality biphasic separator is intuitive and has demonstrated a good performance after being tested by engineering students on their computers. However, the use of the keyboard as interface for the operation of the virtual biphasic separator is the main disadvantage, because several commands are required for the manipulation of the virtual industrial process. In this context, in our future work, we will examine the virtual plant with other devices, such as the Oculus Quest headset, the HTC Pro Virtual Reality System, the HTC VIVE Wireless Adapter, and the VIVE Pro Attachment Kit to improve mobility for industrial oil applications in the field.

**Author Contributions:** Conceptualization, F.F.-B., J.G., J.L. and D.O.-V.; methodology, F.F.-B., J.G., J.L. and D.O.-V.; software, F.F.-B. and J.G.; validation, F.F.-B., J.G., J.L. and P.V.; formal analysis, D.O.-V. and A.N.; resources, P.V. and A.N.; writing—original draft preparation, A.N., J.L., D.O.-V., F.F.-B., J.G. and P.V.; writing—review and editing, A.N., D.O.-V., J.L. and P.V.; visualization, P.V.; project administration, J.L.; funding acquisition, J.L. All authors have read and agreed to the published version of the manuscript.

**Funding:** This research received no external funding.

**Institutional Review Board Statement:** Not applicable.

**Informed Consent Statement:** Not applicable.

**Data Availability Statement:** Not applicable.

**Acknowledgments:** The authors would like to thank the Coorporación Ecuatoriana para el Desarrollo de la Investigación y Academia-CEDIA for their contribution in innovation, through the CEPRA projects, especially the project CEPRA-XIV-2020-08-RVA "Tecnologías Inmersivas Multi-Usuario Orientadas a Sistemas Sinérgicos de Enseñanza-Aprendizaje", as well as the Universidad de las Fuerzas Armadas ESPE and the Research Group ARSI for their support for the development of this work. In addition, the authors would like to thank the ANID BECAS/DOCTORADO NACIONAL under Grant 2019-21190961; and in part by the Secretaría de Educación Superior, Ciencia, Tecnología e Innovación de Ecuador under Grant SENESCYT/ARSEQ-BEC-0058482018.

**Conflicts of Interest:** The authors declare no conflict of interest.

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
