# Peer review of "Development and Application of a Virtual Reality Biphasic Separator as a Learning System for Industrial Process Control"

_electronics, doi:10.3390/electronics11040636_

Round 1

Reviewer 1 Report

The paper describes a Virtual Reality implementation of a Biphasic Separator. It has two purposes: (1) to train students or engineers in process and automatic control and (2) to test control techniques, since the operating  performance of a biphasic separator requires advanced automatic control strategies due to its multivariable and nonlinear characteristics.

Paper is well organized and well written. A few text problems are listed below.

Sections 2 and 2.1 have the same title. Sugestions:
2. Modeling of the Biphasic Separator
2.1. Description of the Biphasic Separator

Authors included the system model in the paper, which should be enough to
replicate their simualtions. Also, the math associated with the controllers design is well described. Although it is hard to follow Section 4 without previous knowledge of control theory.

Software Unity is mentioned several times, it seems to be central to the approach, but it is never explained what Unity actually is, neither it is provided a reference.

"This is achieved by programming a Script where the behavior of the
variables is defined (mathematical model)."
The script runs on Unity ? The mathematical model runs on matlab ? How those thing connect ?

Please provide a diagram showing the runtime components of the system, how they interconect and how they exchange data. Where the plant model
runs ? Where the control strategy runs ? How the VR plataform exchange data with them ?

Equation 36 in Section 4.3: "y" means "and ?  (twice)

Same problem in other paragraphs.

Although the methodology (Figure 2) as a whole seems straightforward, it does have value in setting an approach to use virtual reality in the context of control system design. It integrates two usually very distinct areas.

The authors are careful in providing all the parameters used in the plant simulation and the controllers, allowing for the reproducibility of the experiments. 

---

This sentence seems odd:
"Application in Marine knowledge in a primary school natural science course regarding oysters [7], In a university ..."

In addition, in the training
In addition, they can be used in the training

where researchers such as biofuel characterization
where research such as biofuel characterization

Verifying an operation very similar ...
One can observe an operation very similar ...

Downstream pressure of the gas valve 
downstream pressure of the gas valve 

Downstream pressure of the liquid valve
downstream pressure of the liquid valve

a series of differential Equations are treated
a series of differential equations are treated

y(t + j | t ) it is the optimal
y(t + j | t ) is the optimal

m3/ s,
m^3/s,   (use superscript, several places)

Numerical controller (magenta color),
numerical controller (magenta color),

These are a single setpoint
There are a single setpoint   (?)

a very realistic immersive virtual
a realistic immersive virtual

Author Response

Dear reviewer, we want to thank you for the detailed review and all the observations made. We appreciate your comments which are very helpful in helping us improve the quality of our manuscript. We have tried our best to address all the comments and questions. Please see the attached file. 

Reviewer 2 Report

Elaborated article is of a high scientific level, which describes in detail the individual components of the separator. Several editorial corrections that should be introduced to the final version of the text have been marked in the text. The only critical remark is the incorrect proportion of the selection of the group evaluating the developed virtual tool. Since the process of regulating the separation itself is complicated, first of all it should be directed to experienced chemical engineers who will assess the correctness of its reaction. The authors of the vast majority of the questionnaires sent to students of technical universities who had no experience with this type of equipment.

Author Response

(The authors gave the same response as above.)

Reviewer 3 Report

The authors have written a good paper on learning in technical education. Sometimes, it jumps a little fast (from model to teaching)

My biggest remaining question is what part of it the students liked, and which parts still was up for improvement. If no such data is available, at least a discussion of further investigation and optimisation proposals would be useful. 

Further description of how the model (figure 2) was implemented in teaching might come in useful. We know the number of student, their degree and so on, but how was the model used, which introduction was given by the lecturer and so on. 

Minor note: the pie-charts are very hard to read. Please update with higher resolution and maybe not e.g. white on yellow for the value within a certain quarter. 

Author Response

(The authors gave the same response as above.)

Reviewer 4 Report

The submitted manuscript covers an interesting and novel topic for the scientific community. Virtual reality is an emerging technology with a great projection in today's society. For this reason, its implementation and effectiveness must be analyzed.

In this work there are several deficiencies that the authors must address to improve the quality of the study. The authors must delve into the theoretical framework on the state of affairs. The reading of the following manuscript is recommended:

https://doi.org/10.21556/edutec.2019.67.1327

On the other hand, they should improve the quality of the images presented.

Finally, after presenting the conclusions, the authors must reflect the limitations found and the future lines of action that they intend to cover after the presentation of this work.

Author Response

(The authors gave the same response as above.)

Round 2

Reviewer 3 Report

The authors have addressed all issues, and the additions make the manuscript overall good and well prepared. However, I believe the authors forgot to address my second comment: "Further description of how the model (figure 2) was implemented in teaching might come in useful. We know the number of students, their degree and so on, but how was the model used, which introduction was given by the lecturer and so on."

I hope the authors can address this as well.

Author Response

We thank the reviewer for this comment. We want to apologize for our carelessness because we did not submit your question response, in round 1.

The answer to your comment is attached.

Reviewer 4 Report

The authors have improved the manuscript following the guidelines and observations made.

Author Response

We thank the reviewer.